

# Mimicking prophage induction in the body: induction in the lab with pH gradients

Taylor Miller-Ensminger[1], Andrea Garretto[1,2], Nicole Stark[3,4] and Catherine Putonti[1,3,5]

[1] Bioinformatics Program, Loyola University of Chicago, Chicago, IL, United States of America
[2] Department of Microbiology & Immunology, University of Michigan—Ann Arbor, Ann Arbor, MI, United States of America
[3] Department of Biology, Loyola University of Chicago, Chicago, IL, United States of America
[4] Department of Biology, Indiana University at Bloomington, Bloomington, IN, United States of America
[5] Department of Microbiology and Immunology, Loyola University of Chicago, Maywood, IL, United States of America

## ABSTRACT

The majority of bacteria within the human body are lysogens, often harboring multiple bacteriophage sequences (prophages) within their genomes. While several different types of environmental stresses can trigger or induce prophages to enter into the lytic cycle, they have yet to be fully explored and understood in the human microbiota. In the laboratory, the most common induction method is the DNA damaging chemical Mitomycin C. Although pH has been listed in the literature as an induction method, it is not widely used. Here, we detail a protocol for prophage induction by culture under different pH conditions. We explored the effects of pH on prophage induction in bacterial isolates from the bladder, where the pH is well documented to vary significantly between individuals as well as between healthy individuals and individuals with urinary tract symptoms or disease. Using this protocol, we successfully induced phages from seven bladder *E. coli* strains. Testing conditions and stressors appropriate to the environment from which a lysogen is isolated may provide insight into community dynamics of the human microbiota.

Corresponding author
Catherine Putonti, cputonti@luc.edu

# INTRODUCTION

Exploration of the human microbiome has uncovered a vast diversity of viral species, particularly bacteriophages (see review *García-López, Pérez-Brocal & Moya, 2019*). Phages are vital members of microbiota; they can transform bacterial communities by predation (*Fuhrman, 1999*; *Suttle, 2005*; *Clokie et al., 2011*) and drive bacterial diversity (*Buckling & Rainey, 2002*; *Rodriguez-Valera et al., 2009*; *Tellier, Moreno-Gámez & Stephan, 2014*; *Koskella & Brockhurst, 2014*). Phages also can protect the epithelia from bacterial infection (*Barr et al., 2013*). Core phage communities have been identified within several niches of the human body, and the disruption of these communities has been associated with certain gastrointestinal (*Wagner et al., 2013*; *Manrique et al., 2016*) and oral (*Ly et al., 2014*)

symptoms/disease. To date, most human virome studies have focused on phages in their lytic form. However, the community of phages replicating through the lysogenic life cycle is likely far greater. The majority of bacteria within the human body are lysogens harboring phage sequences (prophages) within their genomes, including the gut (*Breitbart et al., 2003*; *Reyes et al., 2010*; *Minot et al., 2011*), the oral cavity (*Pride et al., 2012*), the skin (*Hannigan et al., 2015*), and the urinary tract (*Miller-Ensminger et al., 2018*). Similar observations have been made in the microbiota of other organisms, e.g., the murine gut (*Kim & Bae, 2018*) and the bee hindgut (*Bonilla-Rosso et al., 2020*). Recent evidence suggests that the lysogenic life cycle may be the dominant life cycle for phages in many environments, not just niches of the body (*Knowles et al., 2016*).

Induction is the process of a prophage leaving the lysogenic cycle and entering into the lytic cycle. A variety of environmental stressors can trigger this process, including: the SOS pathway, reactive oxygen species, etc., although this process can also occur "spontaneously" (see review *Nanda, Thormann & Frunzke, 2015*). In the laboratory, the most commonly used method for induction is Mitomycin C (see protocol *Raya & H'bert, 2009*). Other common laboratory techniques include UV (e.g., *Barnhart, Cox & Jett, 1976*; *Zhang et al., 2019*), antibiotics (e.g., *Shaikh & Tarr, 2003*; *Goerke, Köller & Wolz, 2006*; *Loś et al., 2009*), and $H_2O_2$ (e.g., *Figueroa-Bossi & Bossi, 1999*; *Martín et al., 2009*; *Loś et al., 2009*; *Loś et al., 2010*). A few studies have also employed changes in temperature (e.g., *Kirby, Jacob & Goldthwait, 1967*; *Meijer et al., 1998*; *Lunde et al., 2005*; *Chu et al., 2011*), nutrient availability (e.g., *Wilson, Turner & Mann, 1998*; *Lunde et al., 2005*; *Williamson & Paul, 2006*; *Zeaki, Rådström & Schelin, 2015*), and pH (e.g., *Meijer et al., 1998*; *Lunde et al., 2005*; *Wallin-Carlquist et al., 2010*; *Choi, Kotay & Goel, 2010*). With respect to the human body, efforts have begun focusing on chemicals and conditions relevant to the niche from which the lysogens were isolated (e.g., *Pavlova & Tao, 2000*; *Oh et al., 2019*). In a recent study, food, chemical additives, and plant extracts were shown to induce phages from bacterial taxa common in the human gut (*Boling et al., 2020*). Similarly, diet has been linked to induction in the murine gut (*Kim & Bae, 2018*). While we can speculate that induction occurs within the human microbiome, the rate of induction is unknown. Evidence using a mouse model suggests it is likely to be a frequent event in vivo (*Diard et al., 2017*).

pH-mediated induction was of particular interest to us as we study the urinary microbiota and our prior work found that >80% of bacteria in the bladder are lysogens (*Miller-Ensminger et al., 2018*). *Lactobacillus* species, which lower pH by the production of lactic acid (see review *Stapleton, 2016*), are dominant members of the bladder's microbiome (see reviews *Brubaker & Wolfe, 2017*; *Mueller, Wolfe & Brubaker, 2017*). The pH of urine in healthy individuals can vary widely (*Simerville, Maxted & Pahira, 2005*; *Neugent et al., 2020*), and individuals with Type 2 Diabetes and uric acid stones typically have lower pH values (*Pak et al., 2003*; *Cameron et al., 2006*). Thus, we were interested to see if pH values deviating from the "norm" stress urinary bacteria, thus mediating induction. Upon our review of the literature, we found just a few studies that had explored this environmental factor and vary in the methods employed. Here we present a protocol for pH induction. pH is an important environmental stressor to consider beyond just the urinary microbiota. For instance, pH changes due to abiotic and biotic factors have been reported in soil

(*Obour et al., 2017*) and marine and freshwaters (*Griffith & Gobler, 2020*). As a proof-of-concept, we investigated pH-mediated induction in *E. coli* isolates from the bladder.

## MATERIALS AND METHODS

### Reagents

- Lysogeny broth (LB) (10.0 g/L Tryptone (Cat. No. 97063-390; VWR), 5.0 g/L Yeast Extract (Cat. No. 97064-368; VWR), 10.0 g/L Sodium Chloride (NaCl) (Cat. No. BP358-1; Fisher Scientific))
- Agar (Cat. No. 97064-336; VWR)
- Hydrochloric Acid (HCl) (Cat. No. SA49; Fisher Scientific)
- Sodium Hydroxide (NaOH) (Cat. No. SS266-1; Fisher Scientific)
- Nuclease free water (Cat. No. 97062-790; VWR)
- OPTIZYME$^{TM}$ DNAse I kit (Cat. No. BP81071; Fisher Scientific)
- Promega$^{TM}$ GoTaq$^{TM}$ Flexi DNA Polymerase kit (Cat. No. PRM8295; Fisher Scientific)

### Equipment

- Incubator set to 37 °C. (Shaking at 160 rpm for *E. coli* liquid culture growth.)
- Microcentrifuge
- Disrupter Genie® Cell Disruptor Homogenizer (Cat. No. 89202-318; VWR) (alternatively vortex)
- Water bath
- Thermal cycler
- Petri dishes (Cat. No. 25384-342; VWR)
- P1000 and P10 pipettors and pipette tips
- Serological pipettor and 5 mL serological pipettes (Cat. No. 82050-478; VWR)
- 1 μL sterile loops (Cat. No. 12000-808; VWR)
- 1.7 mL microcentrifuge tubes (Cat. No. 20170; VWR)
- 13 × 100 culture tubes (Cat. No. 47729-572; VWR)
- PCR tubes (Cat. No. 83009-684; VWR)
- 0.22 μm sterile syringe filters (Cat. No. 28145-477; VWR)
- 3 mL sterile Luer tip syringes (Cat. No. 82002-278; VWR)
- pH meter or pH paper (Cat. No. 60786-028; VWR)
- 500 mL media storage bottle

### Reagent setup

LB broth—sterilize via autoclave.
  - Adjust pH to 2, 4, 7, 9, and 11 using HCl/NaOH
LB soft agar (LBSA)—0.7 g agar added to 1 L LB. Mix, autoclave, and store as solid at room temperature.
LB agar plates—15 g agar added to 1 L LB

### Protocol

The protocol and related reagents listed here are specific for induction of *E. coli* prophages. The media listed can be substituted for media specific to the bacterium of study. Likewise,

the incubator conditions and duration of incubation can be adjusted for other bacteria. Furthermore, the protocol listed here often includes overnight (~18 h) incubations; other durations can be explored to see induction efficiencies in different phases of the bacterial growth.

This protocol also includes identification of induced prophages that have been predicted from genome sequences. If the genome of the isolate has not been sequenced, Steps 1 and 5 will be skipped. For bacteria with a sequenced and assembled genome, identification of induced prophages can be conducted (Step 5). If prophage sequences have already been predicted, proceed to Step 1 ii. If primers have been designed for predicted prophages, proceed to Step 2.

Step 1. Identify putative prophage sequences and design PCR primers

i. Given a bacterial genome sequence (either complete or in contigs), the presence of prophage sequences can be predicted using publicly available tools, e.g., PHASTER (*Arndt et al., 2016*), VirSorter (*Roux et al., 2015*), etc.

ii. For each predicted prophage of interest, PCR primers can be designed using, e.g., Primer-BLAST (*Ye et al., 2012*), Primer3 (*Untergasser et al., 2012*), etc. Shorter amplicons will reduce the PCR cycling time for quicker results for identification.

*Recommended Task.* The PCR primers can be tested via a colony PCR (or via amplification of DNA extracted from the isolate). This will be repeated as a control PCR in Step 5.

Step 2. Culture bacteria at alternate pH levels

i. Inoculate 6 mL of LB with a single colony of the *E. coli* containing the prophage and incubate overnight (~18 h) at 37 °C with shaking (160 rpm).

ii. Subculture 1 mL of the overnight *E. coli* culture into 3 mL of pH-adjusted LB and incubate overnight (~18 h) at 37 °C. Here we used pH values of 2, 4, 7, 9, and 11 (selected based upon urine pH levels reported in the literature).

Step 3. Lysate isolation and plating

i. Grow 1 mL of each susceptible host bacterium (here we used *E. coli* B, C, and K-12). Inoculate 1 mL of LB with a single colony and incubate overnight (~18 h) at 37 °C with shaking (160 rpm).

ii. Microwave LBSA until melted. Before use, allow LBSA to cool in a 65 °C water bath or until warm to the touch.

iii. Per susceptible *E. coli* strain to be tested, aliquot 3 mL of LBSA in a glass test tube. When the tube is warm to the touch (but not solidified), add 900 μL of the overnight susceptible *E. coli* culture and then pour evenly on top of an LB agar plate. (Note: for different bacteria, the volume of culture may be adjusted.)

iv. Let the plate cool and *E. coli* lawn solidify (~20 min).

vi. Pipet 1 mL of each of the pH-adjusted *E. coli* cultures into a sterile microcentrifuge tube. Centrifuge at 10,000 g for 2 min.

vii. Transfer the supernatant to a new, sterile microcentrifuge tube.

viii. Filter the supernatant into a new, sterile microcentrifuge tube using the 0.22 μm sterile syringe filters.

ix. Spot 10 μL of filtrate on each susceptible *E. coli* lawn prepared. Allow plate to stand for 5–10 min to minimize the likelihood that the filtrate spot will move when transferred to the incubator.

x. Incubate plate overnight (∼18 h) at 37 °C.

Step 4. Lytic phage isolation

i. Pick each lawn clearing using a sterile loop and put into a microcentrifuge tube with 1 mL of LB. Each should be in a separate tube.

ii. Disrupt (or vortex) tube for 10 min. (We use the Disrupter Genie® Cell Disruptor Homogenizer that vortexes at ∼3,000 rpm.)

iii. Centrifuge tube at 10,000 g for 2 min.

iv. Pipet supernatant into new, sterile microcentrifuge tube. (If only one prophage is predicted for the strain, skip to xiv.)

v. For each lawn clearing picked in i, regrow the bacteria (Step 3 i and ii) from which it was picked, e.g., if one was picked from an *E. coli* C lawn, regrow *E. coli* C.

vii. Replate each supernatant from iv as a pour plate. Mix 3 mL of LBSA and 900 μL of the overnight susceptible *E. coli* culture (grown in Step 4 v) and 100 μL of supernatant (collected in Step 4 iv) in a culture tube and pour evenly on top of an LB agar plate.

viii. Let the plate cool and the lawn solidify.

ix. Incubate plate overnight (∼18 h) at 37 °C.

x. If plate has plaques, pick each individual plaque, depositing it into a single tube of 100 μL of nuclease free water. If the plate has many plaques, one may pick a subset of plaques for further processing.

xi. Disrupt (or vortex) tube for 10 min.

xii. Centrifuge tube at 10,000 g for 2 min.

xiii. Pipet supernatant into new, sterile microcentrifuge tube.

xiv. Filter the supernatant using the 0.22 μm sterile syringe filters into a new, sterile microcentrifuge tube.

*Recommended Task:* This step can be repeated from beginning to end to further purify plaques, i.e., reduce the likelihood that more than one phage sequence is in the filtrate produced in xv. Repetition of this step several times is highly recommended. Protocols for plaque purification often suggest three to five rounds of passaging (*Adams, 1959*; *Łobocka et al., 2014*).

Step 5. PCR identification

i. Treat supernatant with DNAse following reagent protocol.

ii. Aliquot 24 μL of DNAsed lysate into a PCR tube and incubate for 10 min at 95 °C.

iii. Using PCR primers designed for the predicted prophage(s), add 25 μL of PCR master mix, 0.5 μL (10 mM concentration) of each primer to the tube from the previous task.

iv. Negative control reaction should include 25 μL of PCR master mix, 0.5 μL (10 mM concentration) of each primer, and 24 μL of nuclease free water.

v. Positive control reaction should include 25 μL of PCR master mix, 0.5 μL (10 mM concentration) of each primer, 24 μL of nuclease free water, and a colony of the *E. coli* containing the predicted prophage.

vi. Run PCR protocol adjusted for the primers and expected amplicon size.

**Table 1  Number of predicted prophages for each of the VirSorter Categories (Cat #) for the 7 bladder *E. coli* strains tested.**

| Strain | Accession # | Cat1 | Cat2 | Cat4 | Cat5 |
|--------|-------------|------|------|------|------|
| UMB0149 | RRWS00000000.1 | 0 | 5 | 0 | 5 |
| UMB0527 | RRWQ00000000.1 | 0 | 4 | 1 | 8 |
| UMB0906 | RRWO00000000.1 | 0 | 1 | 0 | 6 |
| UMB0923 | RRWN00000000.1 | 0 | 1 | 1 | 3 |
| UMB0934 | RRWJ00000000.1 | 0 | 4 | 0 | 5 |
| UMB1160 | RRWF00000000.1 | 1 | 1 | 1 | 4 |
| UMB1335 | RRVS00000000.1 | 0 | 0 | 1 | 3 |

vii. Run PCR reaction product through a 1.2% agarose gel to confirm amplification.

*Recommended Control:* It is recommended that a 16S amplification is also conducted for the DNAsed lysate to confirm that bacterial DNA is not present in the lysate (and potentially causing a false positive result). 16S primers, e.g., 27f/1492r (*Frank et al., 2008*) and 63f/1387r (*Marchesi et al., 1998*), can be used. Note, the prophage primers can be designed such that their thermal cycling protocol is the same as the 16S primers used (i.e., both reactions can be run using the same protocol) or the 16S PCR can be run in a separate cycle.

## Proof-of-concept assay details

We selected seven strains from our collection for testing. These strains were isolated from urine aseptically via transurethral catheter as part of previous IRB-approved studies (*Hilt et al., 2014*; *Pearce et al., 2014*; *Pearce et al., 2015*; *Price et al., 2016*; *Thomas-White et al., 2016*; *Thomas-White et al., 2018*). Stocks of each strain were made using 900 μL 50% glycerol mixed with 900 μL *E. coli* cells, and stored at −80 °C. Their genomes are publicly available (listed in Table 1) and were predicted to contain prophages by VirSorter (*Roux et al., 2015*). PCR primers were designed for each predicted prophage sequence using Primer3 (*Untergasser et al., 2012*) and synthesized by Eurofins Genomics LLC (Louisville, KY). Table S1 lists the predicted prophages and the PCR primer sequences used. Table S2 lists the thermal cycling conditions used. Primers were tested against the bacteria via colony PCR to confirm that they produced the expected size amplicon. Three laboratory *E. coli* host strains were used as "susceptible hosts"—*E. coli* B (Cat. No.11303; ATCC), *E. coli* C (Cat. No. 8739; ATCC), and *E. coli* K-12 (Cat. No. 25404; ATCC). Cell density of cultures was measured after overnight growth (∼18 h) using a spectrophotometer, set to measure density at 600 λ. Transmission Electron Microscopy was performed for lysates (collected at the conclusion of Step 4). Carbon-formvar coated grids (Electron Microscopy Sciences) were negatively-stained with 2% uranyl acetate for visualization with a JEOL 1200 EX Transmission Electron Microscope. An image was taken at 100k magnification.

## RESULTS

The pH protocol detailed in the Methods and represented in Fig. 1 cultures lysogens at different pH values as a means of triggering prophage induction. Lysate is collected from

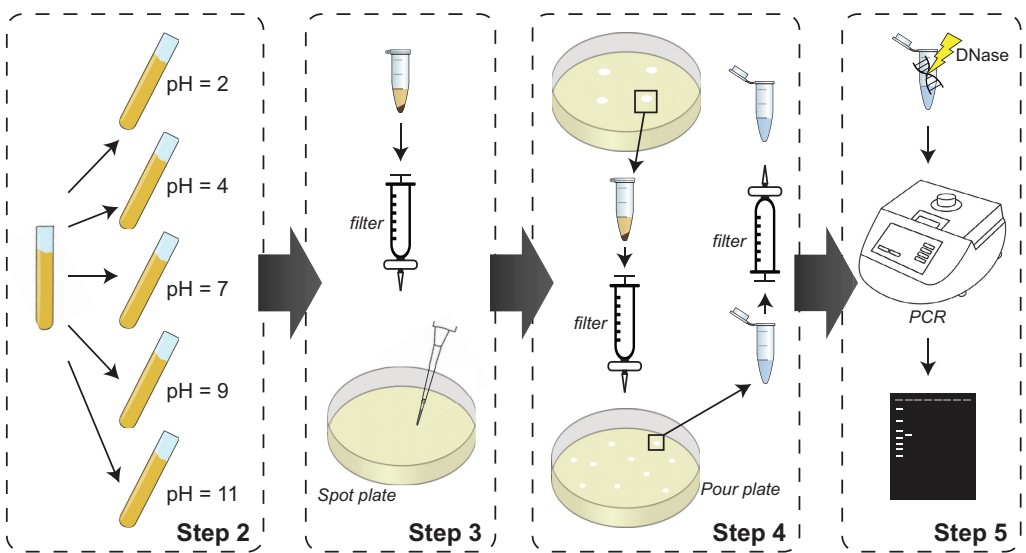

**Figure 1** **Schematic of experimental steps of the protocol from culturing prophage-containing bacteria to PCR-based identification.** Step 1 (not shown) includes prediction of prophage sequences and PCR primer design for use in Step 5. If identification is not necessary, Step 1 and 5 can be omitted; Step 4 will conclude with pure isolates for downstream analysis.

each pH adjusted culture from Step 2, and then filtered and spotted on naïve bacterial lawns in Step 3. In Step 4 of the protocol, lawn clearings are harvested, filtered, and replated with naïve bacteria to identify individual plaques, which are then harvested and filtered again. The protocol includes two optional steps: Step 1, which is not represented in the schematic, and Step 5. These two steps are included for identification of induced phages. Step 1 includes sequencing of lysogen's genome and prophage sequence prediction. Based upon these predicted sequences, PCR primers can be designed for use in Step 5. While the pH conditions shown in Fig. 1 were selected explicitly for our proof-of-concept assays using *E. coli* isolates of the bladder, the methodology described can be fine-tuned for lysogens isolated from other niches. pH varies across sites in the human body (*Evans et al., 1988*; *Lambers et al., 2006*), providing unique growing environments for microbial communities. Additionally, pH changes have been reported in ecological contexts for soil and water environments. Growing isolates at various pH levels that may be experienced in vivo increases our understanding of phage induction.

To test our protocol, we focused on *E. coli* isolates from the urinary microbiome. We tested pH values 2, 4, 7, 9, and 11, thus capturing the range of values recorded for urine samples. The experimental protocol detailed in the 'Methods' section and illustrated in Fig. 1 was implemented for the 7 bladder *E. coli* strains listed in Table 1, each of which is predicted to contain several putative prophage sequences.

Three laboratory *E. coli* strains were used as potential susceptible hosts for induced prophages—*E. coli* B, *E. coli* C, and *E. coli* K-12. Each of the 7 bladder lysogens cultured under each of the 5 pH conditions were spotted onto these three laboratory *E. coli* strains. Of the 105 spots, 49 produced clearings on a bacterial lawn (Table 2) suggesting that pH

**Table 2  Evidence of clearings in the lawns by spot plating.**

| Strain | Observed clearings (*E. coli* B/*E. coli* C/*E. coli* K-12) | | | | |
|---|---|---|---|---|---|
| | pH 2 | pH 4 | pH 7 | pH 9 | pH 11 |
| UMB0149 | -/-/- | +/+/+ | +/+/+ | -/+/- | -/-/- |
| UMB0527 | +/-/+ | +/+/- | +/+/- | +/-/+ | -/-/- |
| UMB0906 | -/-/- | +/+/+ | +/+/+ | +/+/- | -/-/- |
| UMB0923 | -/-/- | +/+/+ | +/+/+ | -/+/- | -/-/- |
| UMB0934 | -/-/- | +/+/+ | +/-/+ | -/-/+ | -/-/- |
| UMB1160 | -/-/- | -/+/+ | -/-/+ | -/-/+ | -/-/- |
| UMB1335 | -/-/+ | +/+/+ | +/+/+ | -/+/+ | -/-/- |

**Notes.**
" +" indicates a clearing was observed. " -" indicates that a clearing was not observed. For each strain and pH tested, the results on the three susceptible hosts are shown with *E. coli* B first, then *E. coli* C, and finally *E. coli* K-12.

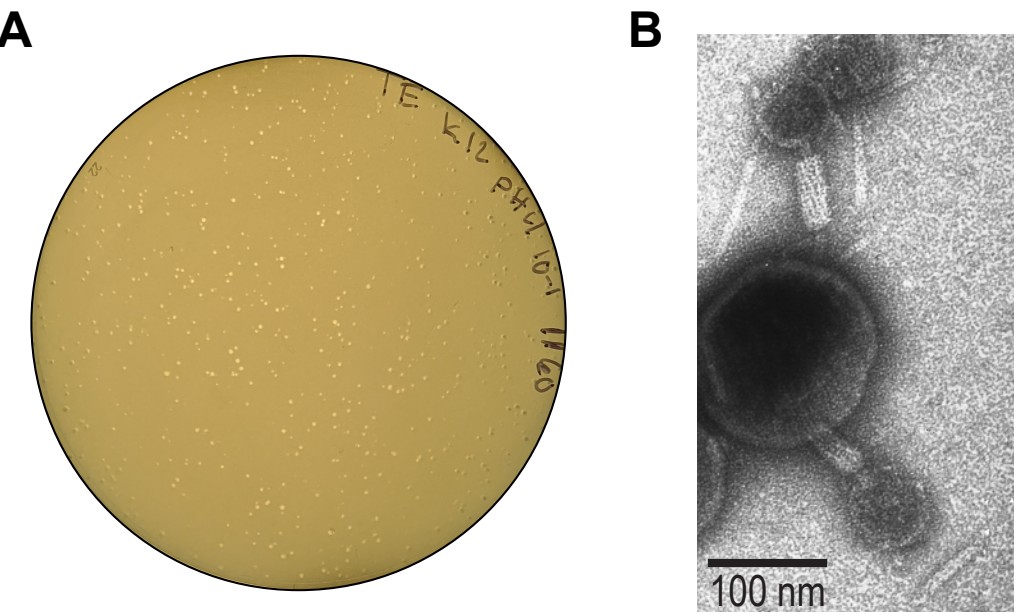

**Figure 2  Evidence of induced phage.** (A) Plaques observed from a pour plate of filtered supernatant (with a 10 × dilution) from UMB1160 grown at pH 4 on the *E. coli* K-12 host. (B) TEM image of the lysate of UMB1160 grown at pH 4. Tailed phages can be seen attached to cell debris.

induced a prophage or spontaneous induction occurred. No clearings were observed on the lawns for the bacterial cultures grown at pH 11, where reduced fecundity of all 7 *E. coli* lysogens was observed (Fig. S1).

Each clearing was isolated, as shown in Fig. 1, and pour plates were prepared. Figure 2A shows one of these pour plates. Here the plate illustrates the filtered supernatant of UMB1160 grown at pH 4 and plated on the *E. coli* K–12 host. Supernatants were confirmed to contain phages via TEM. For example, we observed tailed phages from the lysate of UMB1160 grown at pH 4 (Fig. 2B).
**Table 3** **Phages identified by PCR amplification.** Prophages are named according to the contig in the assembly in which they were found (N #) and the VirSorter Category (Cat #).

| Strain | Prophage | Susceptible host | pH |
|---|---|---|---|
| | N10 Cat5 | *E. coli* C | 4 |
| UMB0149 | N97 Cat2 | *E. coli* B | 4 |
| | N47 Cat2 | *E. coli* C | 7 |
| | N5 Cat4 | *E. coli* C | 4 |
| UMB0527 | N32 Cat5 | *E. coli* C | 4 |
| | N1 Cat5 | *E. coli* B | 4 |
| UMB0906 | N9 Cat5-1 | *E. coli* C | 9 |
| UMB0923 | N3 Cat4 | *E. coli* C | 4 |
| | N3 Cat4 | *E. coli* K-12 | 7 |
| | N5 Cat5 | *E. coli* B | 4 |
| UMB0934 | N5 Cat5 | *E. coli* B | 7 |
| | N39 Cat5 | *E. coli* K-12 | 4 |
| | N39 Cat5 | *E. coli* K-12 | 9 |
| UMB1160 | N12 Cat5 | *E. coli* C | 4 |
| | N11 Cat4 | *E. coli* C | 4 |
| UMB1335 | N11 Cat4 | *E. coli* B | 4 |
| | N11 Cat4 | *E. coli* B | 7 |

To identify induced phages, we performed PCRs in duplicate on individual plaques harvested from the pour plates. Table 3 lists the phages identified. While only 17 of the 49 plaques were identified by PCR, the results highlight two important observations. First, some phages are only induced as a result of a specific pH. For instance, when UMB0149 is grown at pH 7, only one phage, N47 Cat2, is detected. In contrast, when this strain was grown at pH 4, two other phages were detected: N10 Cat5, which lysed *E. coli* C, and N97 Cat2, which lysed *E. coli* B. Second, some phages are induced at multiple pH values. When UMB0923 was cultured at pH 4 and pH 7, N3 Cat4 was detected. Similarly, N5 Cat5 was detected in UMB0934 cultured at both pH 4 and pH 7.

## DISCUSSION

The pH protocol presented here provides a method for inducing prophages in the lab. Based upon our proof-of-concept assays using *E. coli* lysogens, it is quite effective. It is important to note that in patients with a urinary tract infection (UTI) caused be *E. coli*, urine specimens often had a pH around or slightly below 6 (*Lai et al., 2019*). Over half of the pH cultures tested here produced clearings on naïve bacteria lawns. Clearings of bacterial lawns were observed most frequently for *E. coli* isolates grown at pH 4 and 7. Of the 49 observed clearings, we identified 17 phages, matching them to predicted phage sequences via PCR (Table 3). The protocol presented in the Methods can easily be adapted for inducing prophages from other bacterial taxa and isolates from other environments; modifications can include the pH values tested in Step 2 and the culture conditions (temperature, oxygen levels, duration, etc.). These modifications should be made keeping

in mind the conditions from which the lysogen was isolated. Thus, selecting pH values that are likely to cause stress provide insight into prophage induction in vivo.

For our interests, culturing bacterial isolates of the bladder at pH values representative of the broad range typically observed, can mimic one aspect of the bladder environment. While we acknowledge that our results (Table 2) are biased for those induced phages capable of infecting and lysing one of the three laboratory host strains used, propagation through *E. coli* C, B, and K-12 increased the titers of the induced phage and provided visual confirmation of their presence. Inclusion of additional putative hosts may reveal additional phages. Alternatively, the lysate of the pH-altered culture could be directly sequenced thus avoiding the need for a susceptible host bacterium. Furthermore, by performing pour plates, we were able to confirm that the bacterial lawn clearing was the result of phage lysis rather than an antimicrobial produced by the bladder isolate, e.g., a bacteriocin.

Several factors may have contributed to negative PCR results. First, our protocol does not include a complete DNA extraction step, omitted to reduce both cost and time. Rather phage DNA is released by heating the lysate (Step 5 i). It is possible that the phage protein coats were not denatured during this heat treatment. While amending the protocol to include phage lysate DNA extraction may circumvent this issue, the efficacy of DNA extraction protocols varies based on sample type (*Kleiner, Hooper & Duerkop, 2015*; *Castro-Mejía et al., 2015*, p.; *Verbanic et al., 2019*; *Burgener et al., 2020*; *Göller et al., 2020*, p.). Second, the induced phage may have a low burst size, below the detection threshold of standard PCR, recently shown to be 32 copies of the genome per 1 μL input DNA (*Purcell et al., 2016*). Third, the PCR primers may not be targeting the phage genome. Although the PCR primers amplified the bacterial DNA, the bioinformatic prediction of the prophage sequence may be incorrect. Phage sequence predictions tools are constantly improving. It remains a challenge to discern between those prophages capable of entering the lytic cycle and those that cannot. Whole genome sequencing of the lysate produced at the end of Step 4 of the protocol would provide insight into the accuracy of these prophage sequence predictions.

We did not observe clearings on *E. coli* isolates grown at pH 11, and only two isolates grown at pH 2, UMB0527 and UMB1135, produced clearings on the spot plates. At these extreme pH values, we noticed reduced growth of the bacterial cultures (Fig. S1), which may have prohibited prophage induction or phage propagation. While some phages were only identified from a culture for a single pH condition, others were identified in the lysate of cultures from multiple pH levels. This may reflect a range of pH values able to induce the phage or it may simply be the result of spontaneous induction. Spontaneous induction has been observed in isolates from the urinary tract, such as *Streptococcus anginosus* (*Brassil et al., 2020*).

## CONCLUSIONS

The pH protocol is efficient in the induction of prophages from *E. coli* isolates from the bladder. The range of pH values observed in the urine may be the result of differences in microbial community structures and phage induction may be a contributing factor.

Applying stressors that are directly applicable to the urinary tract provides insight into how phages may shape the bladder microbiota. Similar strategies can be employed for other anatomical niches. While here we have focused our efforts on induction by pH, other conditions of the bladder could also be explored. Artificial urine media (e.g., *Brooks & Keevil, 1997*; *Christmas et al., 2002*; *Khan et al., 2017*) are limited in the number of bladder bacterial taxa they can support. As such there are several growth factors that have yet to be determined that may also be triggering or suppressing induction within the urinary microbiota. The protocol presented here considers one well-studied aspect of the bladder environment and provides a means of replicating what might be a natural process in the urinary tract.

## ACKNOWLEDGEMENTS

We would like to thank Bridget Brassil for helping with the TEM image and Laura Maskeri for her assistance with some of the assays. Thanks also to Genevieve Johnson and Jason Shapiro for their feedback on previous versions of the manuscript. For prior patient recruitment, we acknowledge the Loyola Urinary Education and Research Collaborative (LUEREC) and the patients who provided the samples for this study. Thanks also to Dr. Alan J. Wolfe for providing these isolates.

### Funding

This work was supported by the National Science Foundation (Grant 1661357 to Catherine Putonti). Taylor Miller-Ensminger was supported through Loyola University Chicago's Carbon Research Fellowship. The funders had no role in study design, data collection and analysis, decision to publish, or preparation of the manuscript.

### Grant Disclosures

The following grant information was disclosed by the authors:
National Science Foundation: 1661357.
Loyola University Chicago's Carbon Research Fellowship.

### Competing Interests

The authors declare there are no competing interests.

### Author Contributions

- Taylor Miller-Ensminger and Andrea Garretto performed the experiments, analyzed the data, prepared figures and/or tables, authored or reviewed drafts of the paper, and approved the final draft.
- Nicole Stark performed the experiments, authored or reviewed drafts of the paper, and approved the final draft.
- Catherine Putonti conceived and designed the experiments, prepared figures and/or tables, authored or reviewed drafts of the paper, and approved the final draft.

## Data Availability

Data are available at NCBI Assembly Database: RRWS00000000, RRWQ00000000, RRWO00000000, RRWN00000000, RRWJ00000000, RRWF00000000, RRVS00000000.

## Supplemental Information

Supplemental information for this article can be found online at http://dx.doi.org/10.7717/peerj.9718#supplemental-information.

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
