# Peer review of "Mimicking prophage induction in the body: induction in the lab with pH gradients"

_PeerJ, doi:10.7717/peerj.9718_

## Round 0.1 · original submission · Minor Revisions

Both reviewers provided helpful feedback for improving the manuscript and see it as a beneficial contribution to the phage literature.

Reviewer 1 ·

Basic reporting

Mimicking prophage induction in the body: Protocol for induction in the lab with pH gradients by Miller-Ensminger et al is a streamlined proof-of-concept methods description of inducing prophages from lysogens using pH. While the lysogens were isolated from human bladder, it is likely the methods are widely applicable. The method is timely, in that the need to study the influence of prophages (and their lysogens) within microbiomes is rapidly being recognized.

Overall, the writing style is clear and concise. In some areas, however, a few grammatical errors creep in that need to be addressed with a careful read through.
Additional minor improvements are suggested below.

One concept may need clarification: I appreciate the idea that distinct niches provide unique growing environments, including pH, for the microbes that inhabit them. However, in most cases the microbes are adapted to those conditions (e.g., pH) and thus it is likely that deviating from that pH, for example, is stress that mediates the induction. If so, maybe make that more clear to the reader?

Experimental design

Concise. Some clarity needed. For example, In Step 2 (i), please define shaking (what speed?). Orbital shaking speed is often important.

Please also define what 'overnight' means. Does it matter if the bacteria grow for exactly 18hrs? Is the intention to subculture the bacteria while still in log phase or in stationary phase? This may influence induction.

Step 3 (iii), is the LBSA a low melt variety? If so, which one? How low did you bring the temperature down from 65C before mixing and overlaying the soft agar?

Validity of the findings

The findings are clearly reported. The rationale for the study is clear; all data have been provided.

Additional comments

Some additional comments:

Cat is defined in Table 3, but appears first in Table 1.
Table 1 needs a better description. Define "cat" but also make it clear that the number represents the number of predicted prophages.

Table 2 legend: change "platting" to plating.

Figure SF1: needs a legend --when was the OD taken? The legend should emphasize reduced growth/viability of the E coli since it does not do well at the pH extremes.

Table 2: spot assays were repeated 3 times? If so, this notation (+/-/+) means the second experiment failed to produce a clearing?
Spot assays are notoriously problematic, so the reader should be warned. Especially in a situation like this ( -/-/+)-- why did it fail to produce a clearing the first two times?

The differential induction noted in lines 279-281 is an important observation; maybe suggest to the reader followup strategies to quantify this observation?

In line 233, for TEM, was the supernatant concentrated in some way? It would seem the prophage particles would otherwise be too dilute to visualize on the grids.

·

Basic reporting

Overall, this is a well-written paper. It has a very good introduction which is thoroughly referenced. The induction method is clearly described overall and may be very helpful for researchers who wish to avoid mitomycin C either due to its cost or safety issues.

The results and discussion are not as extensive but are acceptable for a methods paper. Key points of the results are discussed. This method is not completely novel, as the authors clearly acknowledge, but a good methods summary is still a useful publication.

There are two minor typos that should be corrected. In line 159, the second use of LBSA is misspelled as “LSBA”. In the legend for Table 2, it should be “plating” not “platting”. Or “spot testing” which is a common way of naming this procedure.

Experimental design

This is not an entirely novel method as the authors state, so it isn’t filling a knowledge gap. But it is filling a procedural gap. Due to the growing popularity of phage therapy, more researchers are beginning to work with phages. Practical methods papers such as this one are worth publishing to help new workers in the field. As well, as the authors note, this method might be effective with other types of bacteria in other environments so this article has the potential to be more broadly applicable. This would have to be tested experimentally of course.

Overall, the methods are very detailed and clear. They should be accessible to experienced and novice phage biologists. There are a few minor revisions needed.

Revision 1. One procedure requires some clarification. In Step 4. ii. (line 178) and Step 4. xi. (line 192), the protocol states that the recovered top agar containing phage is vortexed for 10 minutes. Given that most vortexers are capable of mixing at 10’s of rpms to DNA shearing speeds of 1000’s of rpms, some indication of the shaking speed should be indicated. I presume it is in the 100’s of rpms but this should be stated.

It isn’t clear to me why this step is being performed except perhaps to speed the process of phage release from the agar. In my lab, we typically just leave cored plaques sitting in buffer for 30 minutes to 2 hours with no shaking and get good phage recovery. No change in the paper is needed around this point except for the added detail of rotation rate.

Revision 2. A second detail that should be added is in the optional task in lines 197-198. When trying to obtain a pure strain of bacteriophage from a potential mix, more than one round of passaging is recommended. Adams (Bacteriophages; InterScience: New York, NY, USA, 1959) recommends 3 rounds of passaging while Lobocka and colleagues (The First Step to Bacteriophage Therapy: How to Choose the Correct Phage. In Phage Therapy: Current Research and Applications; Borysowski, J., Miedzybrodzki, R., Gorski, A., Eds.; Caister Academic Press: Norfolk, UK, 2018; pp. 23–67) recommend 5 rounds. In either case, one round is not sufficient especially if the plaque density is high as shown in Figure 2B. This should be noted in the text.

Revision 3. Regarding Figure 2B, if it is possible to replace it with a photograph of a plate with a more dilute sample to give a lower plaque density, this should be done. It would be better if the plaques are distinct and rarely overlapping so that the plaque morphology is clearer. The current picture appears to have multiple plaque types although Table 2 only lists a single phage from the indicated E. coli strain.

Validity of the findings

No additional comments for this area.

Additional comments

This is a nice method. I have students trying to isolate phages right now and mitomycin C was our backup plan if we needed to induce prophages. We may try your method as well.

Fun fact only distantly related to this paper, the FDA recently approved a mitomycin gel for treating urothelial cancer (https://www.fda.gov/news-events/press-announcements/fda-approves-first-therapy-treatment-low-grade-upper-tract-urothelial-cancer) so the types of E. coli used here may be exposed to mitomycin “in the wild” and presumably have phage induction occur. Perhaps urine from patients undergoing this chemotherapy could be a new source of phages.

---

## Round 0.2 · accepted · Accept

Thank you for carefully addressing the reviewer's comments, I am happy to accept this revised manuscript. A few very minor things to double-check in your proofs (line numbers are from the no-track-changes revision)

- Line 211 and 222: Replace "DNased" with "DNase treated"
- Line 276: Consider replacing "fecundity" with "growth" - fecundity isn't often used to describe bacteria
- Line 298: Change "caused be" to "caused by"
- Line 325: There are some problems with reference formatting here (extra p.)